# Mitochondrial DNA Copy Number Is Associated with the Severity of Irritable Bowel Syndrome

**DOI:** 10.3390/medicina60101605

**Published:** 2024-10-01

**Authors:** Soo-Jung Jung, Jae-Ho Lee, Ji-Yeon Lim, Yun-Yi Yang

**Affiliations:** 1Department of Physiology, School of Medicine, Kyungpook National University, Daegu 41944, Republic of Korea; soojung4234@knu.ac.kr; 2Cell and Matrix Research Institute, Kyungpook National University, Daegu 41944, Republic of Korea; 3Department of Anatomy, School of Medicine, Keimyung University, 1095 Dalgubeol-daero, Daegu 42601, Republic of Korea; anato82@dsmc.or.kr; 4Department of Food Science and Nutrition, Keimyung University, 1095 Dalgubeol-daero, Daegu 42601, Republic of Korea; ynyn2121@naver.com; 5Department of Nursing, Healthcare Science & Human Ecology, Dong-Eui University, Busan 47340, Republic of Korea

**Keywords:** irritable bowel syndrome, mitochondrial DNA copy number, mitochondria, mitochondria content

## Abstract

*Background and Objectives*: Irritable bowel syndrome (IBS), as a gastrointestinal disorder, presents with abdomen pain and alterations in the bowel habits. Its pathogenesis remains unclear. Here, we examined mitochondrial DNA copy number (mtCN) in IBS and its clinical value. *Materials and Methods*: mtCN was analyzed in 43 IBS patients using quantitative real-time polymerase chain reaction. Furthermore, data on the clinical characteristics of patients and symptom severity of IBS were collected, and their association with mtCN was analyzed. *Results*: mtCN was higher in patients with IBS (*p* = 0.008) and those with a drinking habit (*p* = 0.004). Smoking and the presence of a sleep partner showed a possible association with mtCN; however, it did not reach significance. The severity of IBS symptoms tended to positively correlate with mtCN (r = 0.279, *p* = 0.070). *Conclusions*: Overall, we demonstrated a potential association between mtCN and the clinicopathologic characteristics of patients with IBS. Further studies considering genetic and clinical factors are required.

## 1. Introduction

Irritable bowel syndrome (IBS) is a common disorder in the gastrointestinal tract, characterized by chronic abdominal pain with constipation, diarrhea, and an altered bowel pattern alternating between constipation and diarrhea, characterized by bloating, incomplete bowel movements, and discomfort [1]. IBS affects approximately 11% of the world population, and 55% of patients are women [2,3]. Its prevalence is 10–25% in the United States, 17–21% in South America, 7–9% in South Asia, and 5.6% in the Middle East and Africa [2,3,4]. In particular, the prevalence rate of IBS among college students in their 20s is higher than that in other age groups [5]. According to a recent study, IBS imposes a social, economic, and psychological burden and is also associated with a low quality of life; the diagnosis and the treatment of IBS are difficult because it affects the visceral–brain axis, which is associated with mental health symptoms such as anxiety and depression [6]. 

Mitochondria generate approximately 90% of adenosine triphosphate (ATP) for cellular energy through the process of oxidative phosphorylation (OXPHOS) in cells. Mitochondria have their own DNA (mtDNA), which encodes many proteins important for assembly and activity, and each mitochondrion contains multiple copies of mtDNA [7]. The total content of mtDNA in cells is regulated according to metabolic requirements [8]. Therefore, mtDNA content is tissue- and developmental-stage-specific and finely regulated by a balance between replication and turnover. Previous studies using human tissues demonstrated that mtDNA copy number (mtCN) per cell may vary by several orders of magnitude: ~1 × 10^5^ mtDNA copies in oocytes [9]; ~4–6 × 10^3^ in the heart; 0.5–2 × 10^3^ in the lungs, liver, and kidney [10]. Although mtDNA plays an essential role in regulating apoptosis and metabolism, it has a high mutation rate [11]. Pathogenic mutations have been documented in mitochondrial tRNA, rRNA, and protein-coding genes, and they invariably compromise mitochondrial gene expression, causing various degrees of OXPHOS deficiency. Mutations occur more frequently in mtDNA than in nuclear DNA (nDNA) because of the insufficient repair systems in the mitochondria, which can lead to various diseases [12]. mtCN in a cell can vary widely by organism, tissue, and cell type. Furthermore, differences in mtCN may be associated with many diseases, especially gastric and colon cancers, and genetic changes are associated with colorectal carcinogenesis [12,13,14].

A previous study has demonstrated that mtDNA levels are reduced in aging and neurodegenerative disorders, like Alzheimer’s disease, Parkinson’s disease, etc. [15]. However, the results differed according to the quality and composition of the specimens. It is not clear how the disruption of the mtCN is related to IBS. To better understand this relationship, we aimed to determine the clinical characteristics of patients with IBS and mtCN and discuss the clinical significance of mtCN in this study. As mtCN may vary depending on age and genetic status, this study was performed in college students without hereditary disorders. 

## 2. Materials and Methods

### 2.1. Participants

Participants who met the inclusion criteria were recruited by advertisements on the bulletin boards in Daegu, South Korea. The inclusion criteria were as follows: (1) college students over 18 years old; (2) meeting the Rome III Diagnostic Criteria for IBS [1]; (3) no previous history of surgeries and diagnosed disorders in the gastrointestinal tract such as obstructive bowel disorders, inflammatory bowel disorders, and lactose malabsorption; (4) no previous history of psychiatric diseases. A total of 43 participants with IBS were contacted. 

The severity of IBS symptoms was measured using previous published criteria [16]. These consist of seven items, and after excluding the categorical questions inquiring about the presence of abdominal pain or abdominal bloating, the remaining five questions were scored 100 points each using the visual analog scale (VAS): Q1 = abdominal pain, Q2 = bloating and flatulence, Q3 = bowel habits (diarrhea and constipation), Q4 = perception of psychological wellbeing, and Q5 = daily life influenced by GI problem. The severity was determined by the sum of the above five scales and was classified as follows: 75–174 = mild, 175–299 = moderate, and 300–500 = severe. Our previous study showed that the reproducibility of this scale at the time of development was stable (85%), and Cronbach’s α level indicating reliability was 0.72 [5].

### 2.2. DNA Extraction 

Blood, to obtain serum, was collected from 43 participants. Informed consent was verbally obtained from all participants and/or their legal guardian. The Institutional Review Board of the Keimyung University Dongsan Medical Center approved the protocols necessary for the research. Genomic DNA was extracted from serum using a DNA extraction kit (Qiagen, Inc., Valencia, CA, USA). Its quantity and quality were measured using NanoDrop 1000 (Thermo Scientific, Wilmington, DE, USA).

### 2.3. Mitochondrial DNA Copy Number

The mtCN was investigated using a real-time quantitative polymerase chain reaction (qPCR) assay. For the quantitative determination of the mtDNA, its contents relative to the nDNA contents were analyzed. Based on a previous study [15], specific primers for the amplification of mitochondrial COX1 and nuclear β-actin were selected. qPCR was performed using the LightCycler 480 II system (Roche Diagnostics, Mannheim, Germany), with a reaction mixture with a total volume of 20 µL, containing 50 ng of DNA, 8 pmol of each primer, and 10 µL of SYBR Green Master MIX (Takara, Kyoto, Japan). The PCR conditions were as follows: 95 °C for 40 s, followed by 40 cycles at 95 °C for 20 s and 60 °C for 30 s.

The threshold cycle number (Ct) values of mitochondrial COXI and β-actin were determined. The level of mtCN in each specimen was normalized against β-actin level in the same participants. All measurements were repeated three times, and five serial dilutions were performed for the control samples.

### 2.4. Statistical Analysis

All statistical analyses were performed using the SPSS statistical package, version 25.0, for Windows. The chi-square test was used to analyze the associations between the variables. A two-tailed *p* value of less than 0.05 was considered to indicate statistical significance.

## 3. Results

In the present study, mtCN was successfully analyzed in all 43 participants. The average mtCN was 0.57 ± 0.3, and the participants were divided into high and low groups according to the median value of mtCN. The clinical characteristics of patients with IBS and mtCN were investigated, and the results are summarized in Table 1. IBS was divided into the following subtypes: diarrhea-predominant, constipation-predominant, and mixed types. Both diarrhea and constipation types tended to be associated with a lower mtCN; however, the association was not significant. mtCN was significantly higher in male participants (*p* = 0.008) and those with a drinking habit (*p* = 0.004). Smoking and the presence of a sleep partner tended to have an association with a higher mtCN but this association did not reach significance (*p* = 0.052). Other clinical characteristics did not have any relationship with mtCN. 

The severity score for IBS symptoms was 319.42 ± 64.84 in this study. Correlation analysis showed that the severity of IBS tended to positively correlate with mtCN (r = 0.279, *p* = 0.070), but the correlation was not significant.

The correlation between VAS scales and mtCN was analyzed, and the results are presented in Table 2. mtCN varied from 0 to about 8, and VAS scales were scored 100 points each. As a result, mtCN showed no significant association with any of the VAS scales. Age negatively correlated with abdominal pain (Q1) (r = −0.329, *p* = 0.031, Figure 1). Abdominal pain was also associated with bloating (Q2) (r = 0.506, *p* = 0.001) and VAS sum (r = 0.380, *p* = 0.012). 

## 4. Discussion

In this study, we confirmed the relationship between mtCN and clinical features of IBS for the first time. As mitochondrial function in IBS is unclear, we studied the relationship between the clinical characteristics of patients with IBS and mtCN. Our study showed that many clinical characteristics are associated with mtCN in IBS. Interestingly, male patients with IBS and patients with a drinking habit had a significantly higher mtCN. Elevated mtCN is considered a marker for oxidative stress and was found to be present in various cancers, though this is still controversial [12,13,14,17]. Smoking and the presence of a sleep partner correlated with a higher mtCN. These factors may influence the quality of sleep [18]. In unmarried college students, the presence of a sleep partner could have various effects on sleep [5]. If your lifestyle and sleeping habits do not match those of your partner, sleep quality deteriorates, causing problems with the rhythm of life and difficulties with bowel habits. A previous study has demonstrated that heavy and frequent smoking and alcohol use are associated with both insomnia symptoms and sleep dissatisfaction [16]. The underlying mechanism of the comorbidity of IBS and insomnia is unclear; the gut–brain axis may have an important role in their relation [19]. A previous study suggested that sleep disorders may occur due to simple digestive disorders such as indigestion [20]. This could cause an imbalance between the autonomic nervous system (ANS) and the hypothalamic–pituitary–adrenal axis, increasing hypersensitivity to visceral pain [20,21,22]. Mitochondrial damage can lead to ATP depletion and excessive Ca^2+^ concentrations, increasing mitochondrial membrane permeability. This enables the accumulation of ROS, contributing to the pathogenesis of a variety of neurological conditions [23]. Our results showed a potential correlation between mtCN and VAS score, including neuro-psychological symptoms, but the correlation was not significant. This finding supports the hypothesis that the mtCN–brain–gut axis may contribute to the development of IBS. However, the detailed molecular mechanism of this should be studied further using in vivo experiment models. 

mtDNA is highly polymorphic and plays an essential role in apoptosis and metabolic regulation. As mtDNA lacks a repair system compared to nDNA, more mutations could occur in mtDNA than in nDNA. Recent studies have shown that single-nucleotide polymorphisms (SNPs) in mtDNA may be associated with the risk of IBS, especially diarrhea-predominant IBS [24,25]. Alterations in mtDNA due to SNPs or other genetic disorders may aggravate IBS symptoms such as pain. Interestingly, regular exercise reduces cell-free mtDNA level and alleviates IBS symptoms [26]. Furthermore, mtDNA SNPs may predict the response to exercise and IBS prognosis in patients. Our results showed that mtCN was positively correlated to IBS severity. These findings suggest that mtCN reflecting mitochondrial DNA changes could be a biomarker for IBS severity or prognosis. However, the effect of mtCN differences and their function is unclear, and further studies should be carried out. 

Caution should be exerted when interpretating our study findings. The number of participants in this study was insufficient, and there is a lack of data on IBS progression. A control group was absent in this study due to ethical reasons. Furthermore, information about this disease is scant, and not many patients regularly visit hospitals for treatment. In this respect, there is low relevance for our hypothesis regarding mtCN’s function as a marker of IBS. Therefore, additional long-term follow-up studies should be conducted with a higher number of patients, including those with psychiatric diseases, such as bipolar disorder, schizophrenia, etc. The functional effect of mtCN was not demonstrated in this study, and its direct effect on IBS symptoms was also unclear. We also examined the correlation between mtCN and each VAS score; however, objective experimental studies of the severity of symptoms should be conducted.

## 5. Conclusions

The present study showed a relationship between the clinical characteristics of patients with IBS and mtCN. Sleep condition factors such as sex, drinking, and smoking, and the severity of IBS symptoms, were associated with mtCN, suggesting mtCN’s potential as a biomarker of IBS. In order to develop a targeted treatment using mtCN for IBS, additional research on the mechanism of IBS pathogenesis is required. Although IBS is not classified as a major disease, continued research is required to improve patients’ quality of life.

## Figures and Tables

**Figure 1 medicina-60-01605-f001:**
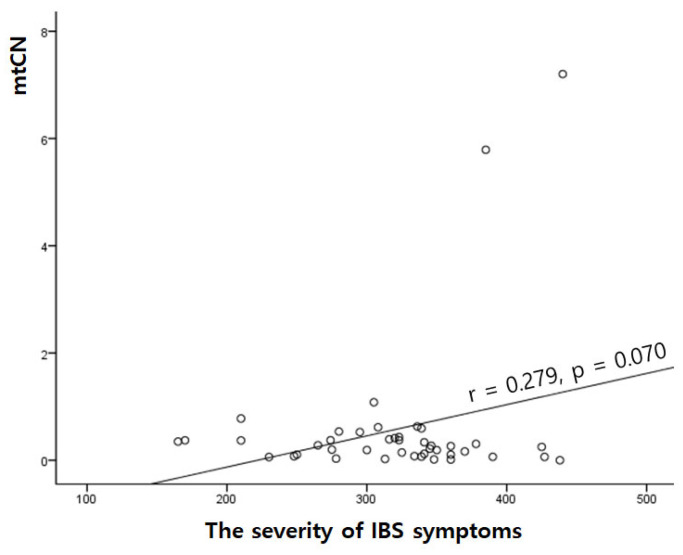
Positive correlation between the severity of IBS symptoms and mtCN.

**Table 1 medicina-60-01605-t001:** Clinical significance of mitochondrial DNA copy number (mtCN) in IBS.

		mtCN (N, %)		*p*
		Low	High	
Subtype	D-predominant	2 (66.7)	1 (33.3)	0.390
	C-predominant	2 (100)	0 (0)	
	Mixed	20 (52.6)	18 (47.4)	
Sex	Male	0 (0)	5 (100)	0.008
	Female	24 (63.2)	14 (36.8)	
Smoking	No	22 (62.9)	13 (37.1)	0.052
	Yes	2 (25.0)	6 (75.0)	
Drinking	No	18 (75.0)	6 (25.0)	0.004
	Yes	6 (31.6)	13 (68.4)	
Sleep partner	No	20 (64.5)	11 (35.5)	0.065
	Yes	4 (33.3)	8 (66.7)	
Night duty	No	16 (53.3)	14 (46.7)	0.619
	Yes	8 (61.5)	5 (38.5)	
Cardiovascular disease	No	24 (57.1)	18 (42.9)	0.255
	Yes	0 (0)	1 (100)	
Inflammatory bowel disease	No	17 (53.1)	15 (46.9)	0.545
	Yes	7 (63.6)	4 (36.4)	
Sleep apnea	No	23 (54.8)	19 (45.2)	0.368
	Yes	1 (100)	0 (0)	
Anxietas tibiarum	No	16 (51.6)	15 (48.4)	0.373
	Yes	8 (66.7)	4 (33.3)	
Visual analog scale	Mild	0 (0)	1 (100)	0.503
	Moderate	7 (53.8)	6 (46.2)	
	Severe	17 (58.6)	12 (41.4)	

D-predominant, diarrhea-predominant; C-predominant, constipation-predominant.

**Table 2 medicina-60-01605-t002:** Correlation between VAS scales and mitochondrial DNA copy number (mtCN).

		mtCN	Age	Q1	Q2	Q3	Q4	Q5	Sum
mtCN	*r*	1	−0.048	0.104	0.232	0.242	0.143	0.030	0.279
	*p*		0.758	0.507	0.135	0.118	0.361	0.847	0.070
Age	R		1	−0.329	−0.132	0.086	0.114	0.270	0.015
	P			0.031	0.398	0.583	0.466	0.080	0.924
Q1	R			1	0.506	−0.070	−0.067	−0.023	0.380
	P				0.001	0.657	0.671	0.883	0.012
Q2	R				1	−0.061	−0.008	0.063	0.521
	P					0.699	0.961	0.686	0.000
Q3	R					1	0.655	0.407	0.709
	P						0.000	0.007	0.000
Q4	R						1	0.286	0.716
	P							0.063	0.000
Q5	R							1	0.535
	P								0.000

Q1 = abdominal pain; Q2 = bloating and flatulence; Q3 = bowel habits (diarrhea and constipation); Q4 = perception of psychological wellbeing; Q5 = daily-life-related GI problem.

## Data Availability

The data presented in this study are available on request from the corresponding author.

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
