# Peer review of "Mitochondrial DNA Copy Number Is Associated with the Severity of Irritable Bowel Syndrome"

_medicina, 2024, doi:10.3390/medicina60101605_

Round 1
Reviewer 1 Report
Comments and Suggestions for Authors
This work describes the search for a link between IBS and mitochondrial DNA number, which is a quite novel approach. The methodology is adequate, although no control group is presented. We understand this may be due to ethical reasons, but the lack of controls undermines the investigation. Results found are tendencys, probably due to the low number of individuals investigated, which makes the results quite weak. Evidence provided is very low to propose mtCN as a marker of IBD, which makes this paper of low relevance.
I considered thos study is preeliminary and needs more work to be published. Authors should increase the number of individuals studied and include a control group. Also, what´s the relvance of studying if patients sleep with someone?
Author Response
Comments 1: This work describes the search for a link between IBS and mitochondrial DNA number, which is a quite novel approach. The methodology is adequate, although no control group is presented. We understand this may be due to ethical reasons, but the lack of controls undermines the investigation. Results found are tendencys, probably due to the low number of individuals investigated, which makes the results quite weak. Evidence provided is very low to propose mtCN as a marker of IBD, which makes this paper of low relevance.
Response 1: Thank you for your kind review and understanding about the low number of individuals. It is our weak point, and it was described in Limitation part. Additional description was added in this part.
Comments 2: I considered thos study is preeliminary and needs more work to be published. Authors should increase the number of individuals studied and include a control group. Also, what´s the relvance of studying if patients sleep with someone?
Response 2: This study included unmarried college students. In these cases, the presence of a sleep partner can have various effects on sleep. If your lifestyle and sleeping habits do not match your partner, sleep quality deteriorates, causing the problems with life rhythm and the difficulties with bowel habits. It was added in Discussion part.
Reviewer 2 Report
Comments and Suggestions for Authors
The article describes an important topic regarding IBS, moreover due to the fact that the pathology is multifactorial and that we don t know the molecular dysregulations that are implicated in IBS development. Although the manuscript is very interesting some points should be more specific.
1. The table 2 which presents the report on mct PCR values and IBS VAS scale should be more detailed in the main text and explained.
2. The primary outcomes of the study should be more specific.
3. How could in the future mDNA represent a biomarker for IBS ? Could you add if mDNA is downregulated in other intestinal pathologies, like IBD or neoplasia from the literature ?
Thank you very much, very nice study topic.
Author Response
The article describes an important topic regarding IBS, moreover due to the fact that the pathology is multifactorial and that we don t know the molecular dysregulations that are implicated in IBS development. Although the manuscript is very interesting some points should be more specific.
--> Thank you for your kind review.
Comments 1: The table 2 which presents the report on mct PCR values and IBS VAS scale should be more detailed in the main text and explained.
Response 1: Detail description about VAS scale was in Method part, 2.1. Participants. For a better understanding, additional description was added in Result part. (The correlation between VAS scales and mtCN was analyzed, and the results are presented in Table 2. mtCN was varied from 0 to about 8, and VAS scales were scored 100 points each. As a result, mtCN had no significant association with any of the VAS scales.)
Comments 2. The primary outcomes of the study should be more specific.
Response 2: In Conclusion section, detail results was added. (The present study showed the relationship between the clinical characteristics of patients with IBS and mtCN. Sleep condition factor such as sex, drinking, and smoking and Severity of IBS Symptoms was associated with mtCN, suggesting the potential of mtCN as a biomarker of IBS.)
Comments 3. How could in the future mDNA represent a biomarker for IBS ? Could you add if mDNA is downregulated in other intestinal pathologies, like IBD or neoplasia from the literature ?
Response 3: In cancer study, increased mtCN was found and it could cause oxidative phosphorylation predicting poor prognosis. This previous study was added in Discussion part. (Elevated mtCN considered a marker for oxidative stress and it was shown in various cancers, though it was still controversial [17]. Smoking and presence of a sleep partner correlated with a higher mtCN. These factors may influence the quality of sleep condition [17].)
Reviewer 3 Report
Comments and Suggestions for Authors
This study only has the feature of the importance of the subject and lacks any quality of study.
The following reasons are important in reviewing the aforementioned study:
1- In choosing the age of the patients, the authors have not specified any report on the age range, and the effect of age on the study is not clear. They have only mentioned the age range above 18 years, the exact determination of the age range, for example, 18 to 40 years or 40 to 65 years and 65 years and above according to the guidelines can help to diagnose the genetic connection of IBS and count the number of mitochondrial DNA copies in the first place, that is, the influence of genes and genetic changes.
It is possible at the age of less than 18 years or 18 to 40 years, we are faced with hereditary changes in the alleles of the patient's genome, or in other words, we have patients in this age range who definitely have IBS disorders and IBS severity, while in the older age range, it is possible to have IBS that occurred in childhood (hereditary disorder) It does not involve the patient, it shows less severity at an older age, and therefore the number of mitochondrial DNA copies can be related to age, heredity, and for this reason, although the correlation is clear in the study, but the correlation is not significant, because the study lacks the quality of detailed investigation.
2- One of the reasons for the poor quality of the study and the lack of detailed examination of the problem can be related to the investigation of mitochondrial gene polymorphisms or SNPs, mitochondrial polymorphism gene changes related to IBS have been widely investigated in previous studies, these changes has been reported abundantly in RNA genes, tRNA genes and mitochondrial core genes, no investigation on mitochondrial DNA polymorphism gene changes has been done in this study and previous studies have been mentioned only in the conclusion section. In general, I should announce that when it is not clear what mutation happened in which part of DNA? This mutation has increased or decreased the expression of which proteins? Does the protein decrease or increase that we need to count its DNA copies? Which protein is increased or decreased causing the symptoms associated with IBS?
So how is the amount of DNA copy number measured to interpret the severity and disorder of the disease!?
3- Irritable bowel syndrome (IBS) is accompanied by many symptoms and problems related to the disease, sleep disorder, diarrhea, abdominal pain, constipation, anxiety, depression, etc., these symptoms do not need to be confirmed, the importance of the issue is determined when the cause of the relationship with these disorders should be investigated, for example, IBS may be related to people who have mental illnesses, bipolar, schizophrenia, etc., and the cause should be investigated, in fact, the symptoms related to IBS are under the control of the intestinal axis and the brain . May be symptoms related to other diseases that lead to IBS.
A quality study on this issue should have more originality and stronger investigations, and this study lacks the quality of appropriate and accurate investigations.
According to the mentioned cases, when we notice the existence of a correlation of a disease and its symptoms with its genetic origin through a superficial examination in such studies, but the correlation relationship is not significant, in fact, the observance of the principles of the investigations and the weakness of the study are evident.
In the end, despite the acceptance of this study due to the importance of the subject, I hope that quality studies will enter the MDPI publications acceptance system.
Thank you
Author Response
This study only has the feature of the importance of the subject and lacks any quality of study.
The following reasons are important in reviewing the aforementioned study:
Comments 1: In choosing the age of the patients, the authors have not specified any report on the age range, and the effect of age on the study is not clear. They have only mentioned the age range above 18 years, the exact determination of the age range, for example, 18 to 40 years or 40 to 65 years and 65 years and above according to the guidelines can help to diagnose the genetic connection of IBS and count the number of mitochondrial DNA copies in the first place, that is, the influence of genes and genetic changes.
It is possible at the age of less than 18 years or 18 to 40 years, we are faced with hereditary changes in the alleles of the patient's genome, or in other words, we have patients in this age range who definitely have IBS disorders and IBS severity, while in the older age range, it is possible to have IBS that occurred in childhood (hereditary disorder) It does not involve the patient, it shows less severity at an older age, and therefore the number of mitochondrial DNA copies can be related to age, heredity, and for this reason, although the correlation is clear in the study, but the correlation is not significant, because the study lacks the quality of detailed investigation.
Response 1: The exact cause of IBS isn't known. IBS prevalence among college students is higher than other age groups because of ill irregular sleep pattern. Therefore, we selected this age of the patients. As your comments, mtCN was varied according to age. To control for these variables, we conducted the experiment in a group of participants with a certain age and no hereditary disorder. This description was added in Introduction.
Comments 2: One of the reasons for the poor quality of the study and the lack of detailed examination of the problem can be related to the investigation of mitochondrial gene polymorphisms or SNPs, mitochondrial polymorphism gene changes related to IBS have been widely investigated in previous studies, these changes has been reported abundantly in RNA genes, tRNA genes and mitochondrial core genes, no investigation on mitochondrial DNA polymorphism gene changes has been done in this study and previous studies have been mentioned only in the conclusion section. In general, I should announce that when it is not clear what mutation happened in which part of DNA? This mutation has increased or decreased the expression of which proteins? Does the protein decrease or increase that we need to count its DNA copies? Which protein is increased or decreased causing the symptoms associated with IBS?
So how is the amount of DNA copy number measured to interpret the severity and disorder of the disease!?
Response 2: There were many mitochondrial polymorphisms, and some SNPs were more frequently found in IBS case compared to control group. Here, we focused the relation between clinical characteristics and disease severity of IBS and mtCN. mtCN change may increase or decrease some proteins associated with IBS. Because the detailed mechanism for this has not yet been revealed, it is difficult to conduct additional experiment to prove this relationship. This description was added in Discussion part.
Comments 3: Irritable bowel syndrome (IBS) is accompanied by many symptoms and problems related to the disease, sleep disorder, diarrhea, abdominal pain, constipation, anxiety, depression, etc., these symptoms do not need to be confirmed, the importance of the issue is determined when the cause of the relationship with these disorders should be investigated, for example, IBS may be related to people who have mental illnesses, bipolar, schizophrenia, etc., and the cause should be investigated, in fact, the symptoms related to IBS are under the control of the intestinal axis and the brain . May be symptoms related to other diseases that lead to IBS.
Response 3: IBS may be associated psychiatric diseases, therefore, we also excluded the patients with mental illnesses, bipolar, schizophrenia, and et al. This description was added in Methods part, 2.1. Participants. The relation between mtCN and these psychiatric diseases is important, and it should be performed further.
Comments 4: A quality study on this issue should have more originality and stronger investigations, and this study lacks the quality of appropriate and accurate investigations.
According to the mentioned cases, when we notice the existence of a correlation of a disease and its symptoms with its genetic origin through a superficial examination in such studies, but the correlation relationship is not significant, in fact, the observance of the principles of the investigations and the weakness of the study are evident.
In the end, despite the acceptance of this study due to the importance of the subject, I hope that quality studies will enter the MDPI publications acceptance system.
Thank you
Response 4: The chief significance of this study is that it was the first to be conducted as a preliminary study on IBS and mtCN. Despite of these limitation, we can raise the level and quality of the article through your excellent reviews. Thank you for your review.
Round 2
Reviewer 1 Report
Comments and Suggestions for Authors
The work has been improved and the issues I stated have been addressed. I believe it is now ready for publishing.
Author Response
Comment: The work has been improved and the issues I stated have been addressed. I believe it is now ready for publishing.
Response: Thank you for your kind review.
Reviewer 3 Report
Comments and Suggestions for Authors
I appreciate the authors' responses to the requested corrections and I enjoyed their in-depth responses. As the last corrections, please correct the following points:
1- In line 156, sleep partner and its effect on irritable bowel syndrome are discussed, while none of the references at the end of the article are related to the issue of sleep partner and IBS disease or are included in this sentence.
2-Reference number 26, which deals with the role of exercise and lifestyle relationship with IBS, is not in the text content, please place number 26 in the appropriate place in the text content, otherwise, you should remove source number 26 from the list of references.
This article is suitable for publication after solving the two problems mentioned.
Thank you
Author Response
I appreciate the authors' responses to the requested corrections and I enjoyed their in-depth responses. As the last corrections, please correct the following points:
-->Thank you for your kind review.
Comment 1: In line 156, sleep partner and its effect on irritable bowel syndrome are discussed, while none of the references at the end of the article are related to the issue of sleep partner and IBS disease or are included in this sentence.
Response 1: Thank you for finding the missing part. We included a reference [ref.5] about to sleep partner and IBS.
2-Reference number 26, which deals with the role of exercise and lifestyle relationship with IBS, is not in the text content, please place number 26 in the appropriate place in the text content, otherwise, you should remove source number 26 from the list of references.
Response 1: Thank you for finding the reference error. ref. 17 was added and there was an error in the number of references that followed. It was revised and total 26 references were checked.